# Mechanism of threonine ADP-ribosylation of F-actin by a Tc toxin

Alexander Belyy [1,6], Florian Lindemann [2,6], Daniel Roderer [1,5], Johanna Funk[3], Benjamin Bardiaux [4], Jonas Protze [2], Peter Bieling [3], Hartmut Oschkinat [2,7✉] & Stefan Raunser [1,7✉]

Tc toxins deliver toxic enzymes into host cells by a unique injection mechanism. One of these enzymes is the actin ADP-ribosyltransferase TccC3, whose activity leads to the clustering of the cellular cytoskeleton and ultimately cell death. Here, we show in atomic detail how TccC3 modifies actin. We find that the ADP-ribosyltransferase does not bind to G-actin but interacts with two consecutive actin subunits of F-actin. The binding of TccC3 to F-actin occurs via an induced-fit mechanism that facilitates access of NAD$^+$ to the nucleotide binding pocket. The following nucleophilic substitution reaction results in the transfer of ADP-ribose to threonine-148 of F-actin. We demonstrate that this site-specific modification of F-actin prevents its interaction with depolymerization factors, such as cofilin, which impairs actin network turnover and leads to steady actin polymerization. Our findings reveal in atomic detail a mechanism of action of a bacterial toxin through specific targeting and modification of F-actin.

[1] Department of Structural Biochemistry, Max Planck Institute of Molecular Physiology, Otto-Hahn-Str. 11, 44227 Dortmund, Germany. [2] Leibniz-Forschungsinstitut für Molekulare Pharmakologie, Robert-Rössle-Str. 10, 13125 Berlin, Germany. [3] Department of Systemic Cell Biology, Max Planck Institute of Molecular Physiology, Otto-Hahn-Str. 11, 44227 Dortmund, Germany. [4] Institut Pasteur, Université Paris Cité, CNRS UMR3528, Structural Bioinformatics Unit, 25-28 Rue du Docteur Roux, F-75015 Paris, France. [5] Present address: Leibniz-Forschungsinstitut für Molekulare Pharmakologie, Robert-Rössle-Str. 10, 13125 Berlin, Germany. [6] The authors contributed equally: Alexander Belyy, Florian Lindemann. [7] These authors jointly supervised this work: Hartmut Oschkinat, Stefan Raunser. ✉email: oschkinat@fmp-berlin.de; stefan.raunser@mpi-dortmund.mpg.de

Entomopathogenic bacteria from the genera *Photorhabdus*, *Xenorhabdus* and *Serratia* have long stirred interest for their potential in pest control[1,2]. Among other virulence factors with extreme potency against insect cells, these bacteria produce a family of toxins called toxin complexes (Tc)[3]. Genes encoding Tc toxins were also found in the genomes of the human pathogens *Yersinia pseudotuberculosis* and *Yersinia pestis*, suggesting their involvement in human diseases[4].

Tc toxins consist of three subunits (TcA, TcB, and TcC) (for review[5]). TcA forms a large, homo-pentameric >1 MDa bell-like structure. It is composed of a central channel surrounded by a shell with receptor-binding domains at its periphery[6–8]. TcB and TcC together form a cocoon of about 300 kDa that encapsulates the actual cytotoxic enzyme of about 30 kDa resulting from the C-terminal autoproteolytic cleavage of TcC[6,9]. While most of the Tc toxin machinery is conserved, the cytotoxic enzyme varies largely between bacterial strains and is therefore referred to as hypervariable region (HVR)[10]. The HVR can be replaced by other small proteins to turn Tc toxins into customizable molecular syringes for delivery of molecules of interest[11].

Previously, our group described several crucial steps of the intoxication process. We demonstrated that upon formation of the holotoxin, high-affinity interaction between TcA and TcB triggers a conformational change in the six-bladed β-propeller domain of TcB, which opens the cocoon and initiates translocation of the HVR into the channel[12]. Once secreted by the bacteria, the fully assembled toxin interacts with the target cell surface glycans and glycosylated cell receptor(s)[13], initiating endocytosis. Acidification of the Tc toxin-containing endosome triggers the prepore-to-pore transition and the release of the cytotoxic effector into the cytoplasm of the target cell[14,15]. There, the effector adopts its active conformation either spontaneously[11] or additionally assisted by the host cell chaperones[16]. Finally, the enzymatic activity of the effector interferes with essential cellular processes, ultimately leading to cell death.

Hundreds of HVRs have been predicted bioinformatically[17], but only two have so far been characterized, namely the ones of TccC3 and TccC5 from *Photorhabdus luminescens*[18]. Both proteins belong to the large family of ADP-ribosyltransferases (ARTs), which includes a wide range of toxins from human pathogens, including *Bordetella pertussis*, *Corynebacterium diphtheriae*, *Clostridium botulinum* and *Vibrio cholerae*[19]. Based on the conservation of functionally critical residues, these enzymes are additionally classified into H-Y-E or R-S-E clades. However, some toxins do not belong to either clade. All members of the ADP-ribosyltransferase family perform a specific post-translational modification: they hydrolyze NAD$^+$ into nicotinamide and ADP-ribose and transfer the latter to the target host cell proteins[20]. While TccC3 HVR ADP-ribosylates actin at T148, TccC5 HVR ADP-ribosylates Rho GTPases at E61 and E63. Together, their activities promote aberrant actin polymerization, cause actin clustering and aggregation, and ultimately cell death[18,21]. However, the molecular mechanism of action of these effectors underlying their pathogenicity is poorly understood.

Here, we show that TccC3 HVR modifies only actin filaments (F-actin) and not monomeric actin (G-actin) as previously proposed[18,21]. We present the structures of the TccC3 HVR enzyme in the apo-state, in complex with F-actin, as well as the structure of ADP-ribosylated F-actin, thereby providing three structural snapshots of TcHVR activity. The ADP-ribosylation prevents interaction with cofilin and other depolymerizing factors, impairing actin network turnover. Taken together, our results provide insights into the architecture and mechanism of action of effectors secreted by Tc toxins.

## Results

**Structure of TccC3 HVR.** To understand the processing and function of a Tc toxin HVR as well as its mechanism of action, we chose the TccC3 HVR from *Photorhabdus luminescens* (hereafter TcHVR) for our studies. Even before the holotoxin consisting of TcA, TcB and TcC is assembled, the TcHVR effector domain has already been autoproteolytically cleaved and resides inside the TcB-TcC cocoon. TcHVR is presumably in an unfolded or partially unfolded state as suggested by previous X-ray and cryo-EM structures in which TcHVR was not resolved[6,9,12,14]. Cross-linking mass spectrometry revealed that while the N-terminus of the ADP-ribosyltransferase resides close to its cleavage site, the rest of the protein assumes random orientations, indicating that the position of the protein is variable inside the cocoon[14]. However, it remains unanswered whether TcHVR is also unfolded. We therefore recorded $^1$H-$^{15}$N correlations of the TcB-TcC cocoon (2474 amino acids) with (Fig. 1a) and also without toxin by ultrafast magic-angle-spinning (MAS) NMR, and compared the first spectrum to the assigned solution $^1$H-$^{15}$N correlation of TcHVR alone (Fig. 1a) that contains characteristic signals with extreme chemical shifts from the folded core of TcHVR. Their putative locations in the MAS NMR spectrum of the TcB-TcC complex are marked by circles in Fig. 1a. If the protein is folded inside the cocoon, cross peaks with very similar chemical shifts are expected to appear inside these circles in the blue cocoon spectrum for buried residues of the effector. Since such signals are not observed, we conclude that the effector is not folded. To test whether all expected signals are observed, we estimated the number of arginine N$_\varepsilon$H signals detected in the $^{15}$N chemical shift region between 80–90 ppm. Integration of spectral areas with sets of well-defined arginine N$_\varepsilon$H cross peaks (Supplementary Fig. 1 and Supplementary Table 1) and subsequent comparison to integrals of all other relevant spectral areas yielded estimates of 140–150 cross peaks, consistent with 150 arginine residues present in TcB-TcC.

Once TcHVR has passed the translocation channel of TcA, the protein folds either spontaneously or assisted by chaperones in the host cell cytoplasm[11,22]. To study TcHVR (282 amino acids) both functionally and structurally, we recombinantly produced the protein in *E. coli* and purified it to homogeneity. We then performed numerous crystallization trials but irrespective of protein construct boundaries or conditions chosen, we could not obtain diffracting crystals. We therefore determined the structure of TcHVR in the absence of NAD$^+$ by NMR in solution, revealing a long, flexible N-terminus (residues 1 – 101, Supplementary Fig. 2) and an ART domain (TcART) spanning residues 102 - 281 (Fig. 1b). TcART comprises eight antiparallel β-strands (Fig. 1c) of which the perpendicularly oriented triads β1/3/8 and β2/7/6 are connected by two hydrogen bonds between Y112 in β1 and A195 at the end of β2 as anchoring points (Supplementary Fig. 3b). These essential β-sheet segments follow the overall organization of the conserved ART domain fold[23] (Fig. 1c and Supplementary Fig. 3a, c), including conservation of the two characteristic Y112-A195 hydrogen bonds whose equivalents are present in most ART structures (see e.g. PDB codes 1GIQ, 5ZJ5, 4Z9D, 4TLV, 1PTO, 1WFX, 6E3A, 6RO0). However, of the four α-helices in the conservation pattern only α1 is observed at its expected position between β1 and β2. In place of a usually conserved second α-helix adjacent to the binding pocket, the segment 200–208 is disordered according to relaxation measurements (Supplementary Fig. 4). As a unique feature of TcART, α2, α3 and α4 form an exposed bundle inserted between α1 and β2 (Fig. 1b, c). R113 in β1, S193 in β2 and E265 at the beginning of β7 represent the R-S-E motif (Fig. 1d). Of the non-canonical features for ARTs, strands β1 and β3 are much longer than the other strands and strongly twisted. Together with

strand β8 they form an extended arch-shaped β-sheet that reaches over the second, also twisted β-sheet formed by β2, β5, β6 and β7, thereby forming a roll-like structure.

Aiming at a model of the NAD$^+$-bound TcART, we investigated the complex by NMR. However, distance-dependent cross peaks between protein and ligand were not observed in NOESY-type spectra, presumably due to unfavorable exchange conditions. Therefore, we performed NMR chemical shift titrations employing $^1$H-$^{15}$N HSQC spectroscopy to monitor ligand-induced chemical shift changes. The largest effects on NH chemical shifts (Supplementary Fig. 5) were observed for Y112,

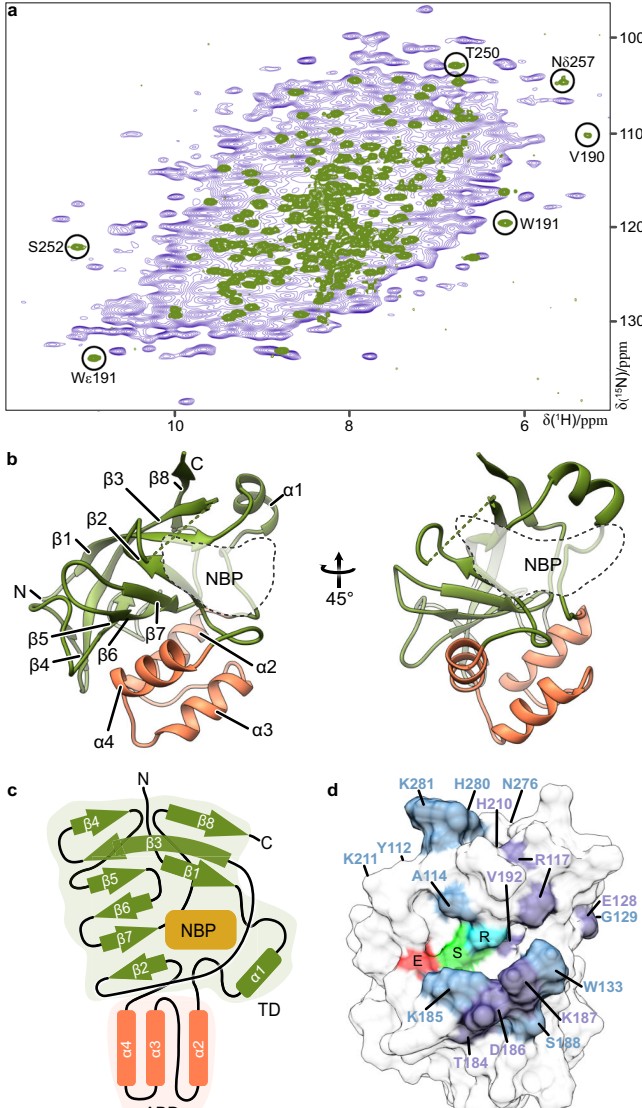

**Fig. 1 Structure of TcART. a** Overlay of two-dimensional $^1$H-$^{15}$N correlation spectra of the TcB-TcC cocoon (blue) and TcHVR (green), with signals diagnostic for folded TcART indicated by circles. **b** Atomic model of an NMR structure of TcART that is closest to the average. NBP: nucleotide-binding pocket. **c** Schematic representation of its secondary structure. TD: transferase domain, ABD: actin-binding domain **d** Surface model of the view shown to the right in **b**, with the 17 surface residues whose chemical shifts are affected most upon addition of a three-fold concentration of ligand indicated either in light blue or in case of the seven top shifting residues in dark lilac. R, S, and E denote the conserved motif consisting of R113, S193, and E265 shown in cyan, green, and red, respectively. K211, Y112, N276, and G129 are hidden or on the backside. The flexible loop 200–208 is omitted.

A114, R117, the region around E128 and G129, T184-S188, H210 and K211, F268, and the region N276-K281 (Fig. 1d), and for the N$_\epsilon$H signals of R113 and R117. The comparison of our NMR ensemble with structures of bacterial R-S-E-type ARTs in complex with NAD$^+$ [24] from *Clostridium perfringens* (1GIQ), *Streptomyces coelicolor* (5ZJ5), and *Bordetella pertussis* (4Z9D) confirmed that the region with the largest chemical shift changes corresponds indeed to the NAD$^+$ binding cleft, containing the three characteristic ART residues[18] (Fig. 1d, Supplementary Fig. 3). Docking studies on the basis of the solution structure to obtain a model of the NAD$^+$ complex did not lead to satisfying results since too many residues are involved in structural rearrangements as suggested by the chemical shift titration experiment, where e.g. strong responses are observed for the stretch 184–188, including those of K185 whose interaction with E265 blocks the potential binding site (Supplementary Fig. 5).

To find out if the N-terminal flexible region (residues 1–102) of TcHVR is important for the function of the enzyme, we designed a set of fragments with N-terminal deletions, and endogenously expressed them in *Saccharomyces cerevisiae*. Expression of the full-length TcHVR, as well as the variant with a deletion of the 100 N-terminal residues fully hindered cell growth (Supplementary Fig. 6), indicating that the N-terminal domain is indeed not required for the enzymatic activity of the TcHVR. We believe that its main function is to act as linker between the ART and the cocoon to fulfill charge and size requirements of the Tc translocation system[11].

By deleting additional 10 residues (residues 100–110) which are already part of the ART domain, the functionality of the toxin was abolished and yeast growth was restored. The same was observed for a construct missing 10 C-terminal residues (Supplementary Fig. 6a), indicating that the structured region of TcHVR (residues 101–282) comprises the complete functional ART domain.

**Structure of the TcART-F-actin complex**. The clade of R-S-E ARTs includes several toxins that selectively modify monomeric (G-)actin. Prominent examples are *Clostridioides difficile* transferase (CDT)[25], *Clostridium botulinum* C2 toxin[26], *C. perfringens* iota toxin[27], *C. spiroforme* toxin (CST)[28] and the *Bacillus cereus* vegetative insecticidal protein (VIP)[29]. These ARTs target selectively R177, which is only accessible in G-actin. In contrast, TcART modifies actin at T148[18], which is accessible in both monomeric and filamentous (F-) forms of actin. Previously, it was suggested that both forms of actin can be substrates of TcART[21]. However, in that study, actin was not stabilized and could therefore spontaneously polymerize during the reaction. Therefore, to define if TcART prefers G- or F-actin as substrate, we performed an ADP-ribosylation assay stabilizing the actin forms by latrunculin and phalloidin, respectively and compared it to the enzymatic component of the iota toxin. In contrast to iota toxin, which, as expected, modified only G-actin, *P. luminescens* TcART ADP-ribosylated exclusively F-actin (Fig. 2a). This experiment identifies TcART as the bacterial ART that specifically modifies actin filaments.

To understand how TcART interacts with F-actin and ultimately modifies it, we aimed to solve the structure of the TcART-F-actin complex using single-particle cryo-EM. To this end, we expressed a construct comprising only the ART domain (residues 101–282) and we performed a series of experiments to reconstitute and stabilize the complex with F-actin. However, regardless of the presence of NAD$^+$ or the use of F-actin, in which T148 is mutated to asparagine to prevent ADP-ribosylation, the TcART exhibited low affinity to its substrate, making high-resolution analysis

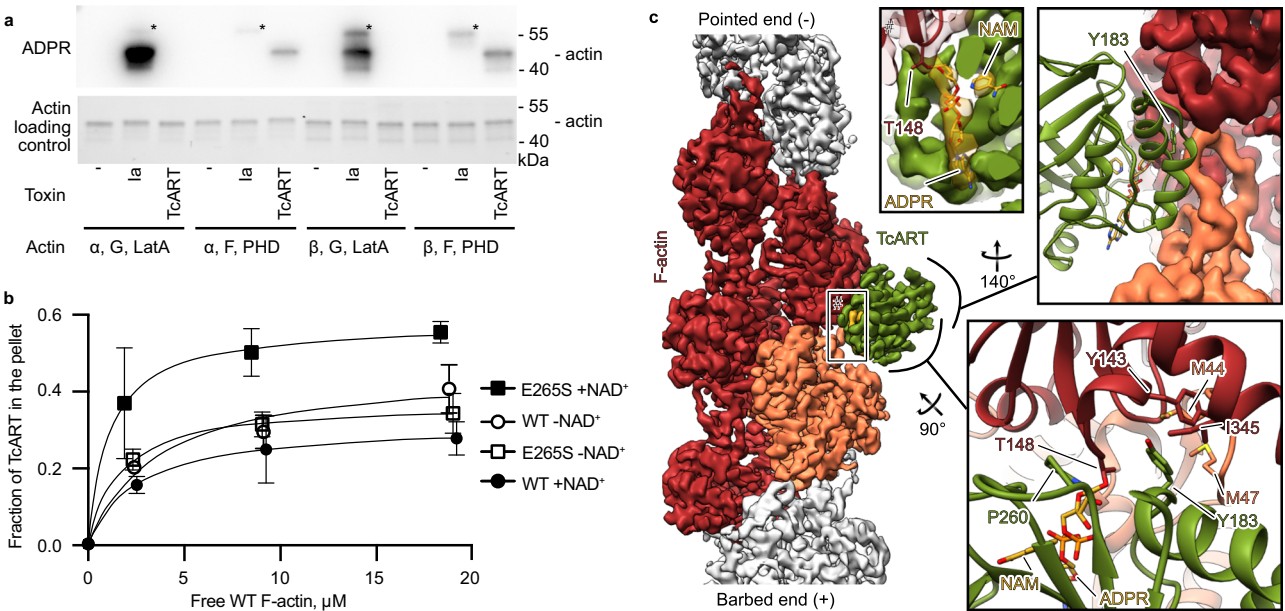

**Fig. 2 TcART interacts with F-actin. a** ADP-ribosylation of 1 µg of rabbit muscle α- or human cytosolic β-actin in G-form stabilized by latrunculin A (LatA) or in F-form stabilized by phalloidin (PHD) by 10 ng of TcART or iota toxin (Ia). The reaction mixture was first separated by SDS-PAGE (lower panel), blotted, and developed with anti-ADPR reagent in western blot (upper panel). Auto-ADP-ribosylation of Ia is marked with an asterisk. The experiment was performed twice, a representative result is shown. **b** The fractions of TcART that cosedimented with phalloidin-stabilized F-actin were quantified by densitometry and plotted against F-actin concentrations. The data are presented as mean values, the error bars correspond to standard deviations of 3 independent experiments. **c** Cryo-EM density and corresponding atomic model of the TcART-F-actin complex (toxin is in green, actin subunits are in orange and red). ADPR – ADP-ribose, NAM – nicotinamide. Uncropped gels and Western blots can be found in Supplementary Fig. 11. Source data are provided as a Source Data file.

difficult (Fig. 2b, Supplementary Fig. 7a, b). Previously, it was demonstrated for the iota toxin that the addition of ethylene glycol stabilizes the iota toxin-G-actin complex by impairing the ADP-ribosylation reaction[30]. However, the addition of high concentrations of ethylene glycol is incompatible with single-particle cryo-EM, because of the dramatically reduced contrast. Instead, we attempted to introduce a mutation in TcART to impair its enzymatic activity in order to stabilize the complex. We chose to mutate E265 to serine, since an equivalent mutation in iota toxin (E380S) was shown to inhibit enzymatic activity[30]. Indeed, enzymatic activity was drastically reduced and complex formation with F-actin was significantly enhanced for this variant (Fig. 2b and Supplementary Fig. 7a, e).

Using this variant, we then performed single-particle cryo-EM analyses and obtained a 3D reconstruction of the complex at an average resolution of 3.8 Å that allowed us to build an atomic model of the involved proteins (Fig. 2c, Supplementary Fig. 8, 9, Supplementary Table 2). The TcART binds with two helices of the three-helix-bundle (α2-α4) like the skids of a helicopter to a surface formed by two consecutive actin subunits (Fig. 2c). In addition, loops extending from the β-sheet roll interact with the upper actin subunit. These interfaces, that are mostly stabilized by electrostatic interactions and salt bridges (Supplementary Fig. 9a), lock the position of the enzyme on the filament and place the catalytic site of the ART close to T148 of actin. The observation that the TcART-F-actin interface involves two actin subunits explains the substrate specificity of the TcART for F-actin. A key residue of TcART within the F-actin interface is Y183, which wedges into a hydrophobic pocket constituted by two adjacent actin subunits (Fig. 2c and Supplementary Fig. 7f, 9b). The mutation of Y183 to serine in TcART dramatically impaired its enzymatic activity in vitro and toxicity in yeast (Supplementary Fig. 7c, e).

In other members of the R-S-E clade of bacterial ARTs[24], the residue (φ) in the third position in the ADP-ribosylating toxin turn-turn (ARTT) loop motif X-X-φ-X-X-E/Q-X-E, forms essential hydrophobic interactions with substrates. Although the corresponding residue (P260) of TcART, sits at the interface to F-actin, it does not interact there with any specific residue or with a hydrophobic patch (Fig. 2c and Supplementary Fig. 7f). Thus, it defines rather the secondary structure of the toxin than playing a significant role in stabilizing the TcART-F-actin interface. Indeed, the TcART variant with the mutation P260G was still active in vitro and toxic in yeast (Supplementary Fig. 7c, e). Thus, the substrate recognition mechanism of TcART differs from other members of the R-S-E clade of bacterial ARTs.

ARTs of the R-S-E clade modify arginine, asparagine, glutamine or threonine. It has previously been suggested that the sixth amino acid in the ARTT loop plays an important role in the target residue specificity as it is presumably involved in the formation of an intermediate state[31]. In the case of the *Bacillus cereus* C3 ART, replacing glutamine with glutamate at this position results in the enzyme modifying asparagine instead of arginine[32]. To test whether this residue is also involved in target residue specificity in TcART, we replaced D263 with asparagine or glutamine and tested whether these mutants can modify F-actin in which T148 has been replaced by asparagine. Our in vitro ADP-ribosylation tests, as well as toxicity assays in yeast, clearly demonstrated that the TcART variants did not modify F-actin T148N and showed only reduced activity towards wild-type F-actin (Supplementary Fig. 7d, g). This indicates that although we cannot change the target residue specificity as in the case of the *Bacillus cereus* C3 ART, D263 plays a central part in the enzymatic reaction and cannot be replaced by other residues. Moreover, the finding that the T148N mutation in actin renders

yeast resistant to TcART indicates that there is no further substrate of the TcART enzyme besides F-actin.

Interestingly, we observed two additional densities in the nucleotide-binding pocket of TcART: one of elongated shape starting from T148 of actin and continuing to the middle of the toxin, and the other one, of spherical shape, in proximity of A114 of the ART (Fig. 2c). We attributed these densities to ADP-ribose and nicotinamide, respectively. While the ADP-ribose is covalently bound to T148 of actin, nicotinamide is located in an adjacent pocket of the ART. Adenosine of the ADP-ribose interacts with the side chains of R117 and W133, and the phosphates are coordinated by R113 and K185 (Fig. 3 and Supplementary Fig. 7h). The nicotinamide is kept in place by

hydrogen bonds with the backbone of A114 (Fig. 3). Our structure of TcART-F-actin demonstrates that the engineered mutation E265S stabilizes the enzyme in the post-reaction state, in which F-actin is ADP-ribosylated.

When we compared the apo TcART structure with its F-actin complex, we found that a number of TcART residues at the interface to F-actin rearrange, indicating an induced-fit mechanism. Interestingly, this conformational change involves K185, which in the apo state forms a salt bridge with E265, blocking the nucleotide-binding pocket (Fig. 3a, b and 5). Thus, upon substrate binding, the K185-E265 gate opens, and the catalytic center becomes easier accessible for $NAD^+$, suggesting that F-actin binding precedes $NAD^+$ binding. This hypothesis is also supported

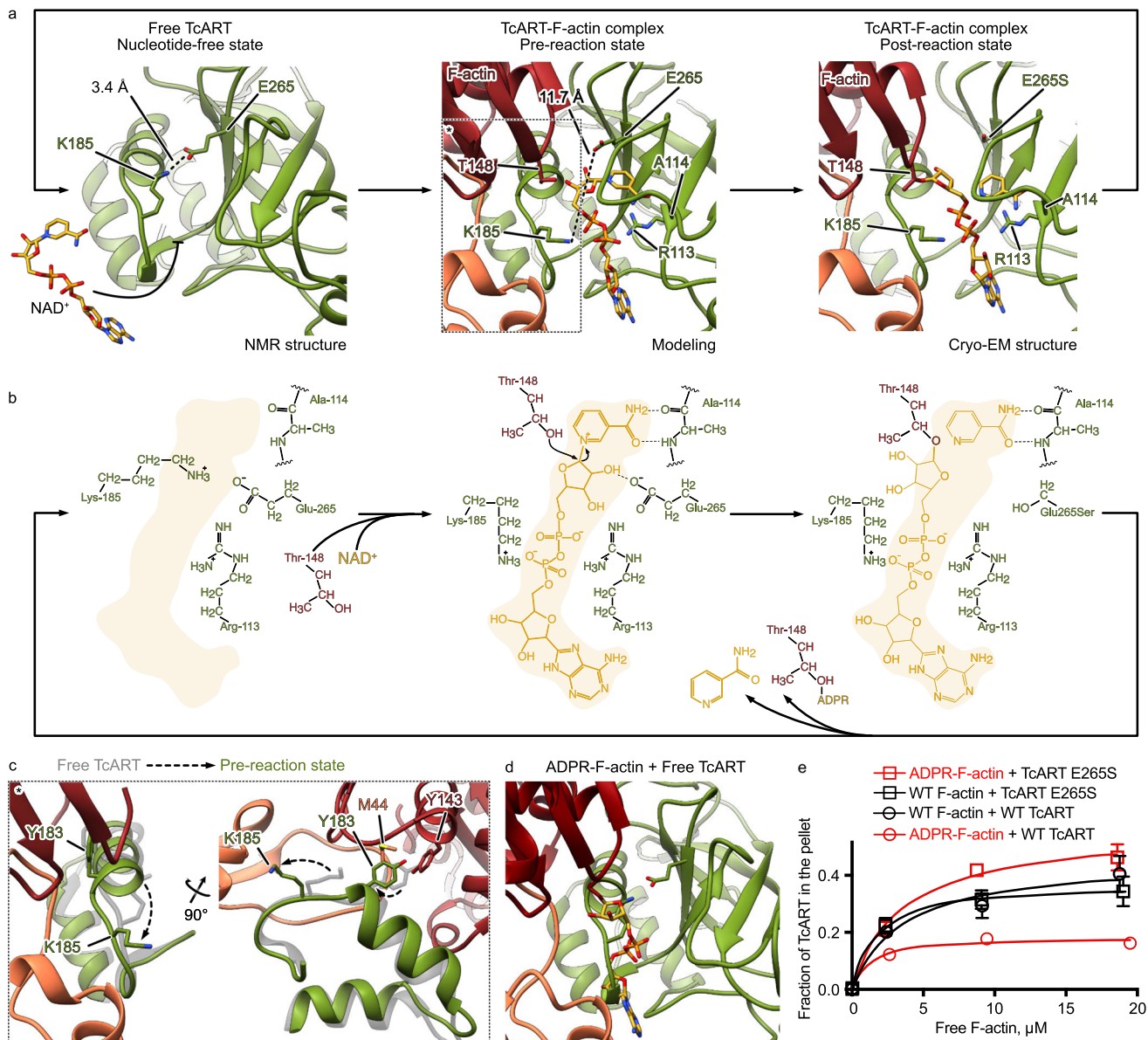

**Fig. 3 Proposed mechanism of ADP-ribosylation by TcART. a** Atomic models of TcART in the apo-state, $NAD^+$-bound TcART-F-actin and the complex of ADP-ribosylated F-actin, TcART and nicotinamide. The $NAD^+$-bound TcART-F-actin structure was modeled based on the cryo-EM structure of the complex of ADP-ribosylated F-actin, TcART and nicotinamide. Note the interaction between K185 and E265 in the apo-state. **b** Schematic representation of the proposed reaction mechanism. **c** K185 relocation during the induced fit interaction between TcART and F-actin. **d** ADP-ribose of modified F-actin overlaps with the binding site of apo TcART on F-actin, preventing the opening of the gate and formation of the complex between TcART and ADP-ribosylated F-actin. **e** Fractions of TcART that cosedimented with phalloidin-stabilized WT or pre-ADP-ribosylated (ADPR-) F-actin were quantified by densitometry and plotted against F-actin concentrations. The data are presented as mean values, the error bars correspond to standard deviations of 3 independent experiments. Source data are provided as a Source Data file.

by our cosedimentation experiments which showed that the affinity of TcART for F-actin is independent of the presence of NAD$^+$, suggesting that F-actin does not prefer TcART-NAD$^+$ over ligand-free TcART (Supplementary Fig. 7a, b).

**Mechanism of ADP-ribosylation by TcART**. To understand the molecular details of the ADP-ribosylation reaction, we computationally mutated S265 back to glutamate and removed the ADP-ribose and nicotinamide from our model of the TcART-F-actin complex in the post-reaction state to obtain a model of the TcART-F-actin complex in the pre-reaction state. We then performed docking experiments with NAD$^+$. Expectedly, the overall coordination of NAD$^+$ is very similar to ADP-ribose and nicotinamide in the post-reaction TcART-F-actin complex, but now E265 of TcART forms a hydrogen bond with the 2'-OH and a salt bridge with the N$^+$ of the nicotinamide riboside (Fig. 3 and Supplementary Fig. 7i). Thus, E265, which is also involved in the gate above the nucleotide-binding pocket in apo TcART, can in principle stabilize an oxocarbenium ion as described in the nucleophilic substitution reaction by iota toxin[30]. If so, the previously engineered mutation E265S could slow down the reaction, allowing us to capture the transient TcART-F-actin complex (Fig. 2).

Comparison of the models with each other and with the NMR apo structure of TcART allows us to describe the ADP-ribosylation in detail (Fig. 3a, b and Supplementary Movie 1). In the pre-reaction state, E265 and A114 fix the nicotinamide-ribose bond exactly in line with the path to the side-chain oxygen of T148 of actin, ultimately enabling a productive nucleophilic attack on NC1 of the N-ribose either by T148 directly, mediated by internal "breathing" motions within the protein complex, or involving the intermediate action of a water molecule, and leading in its course to the formation of a covalent bond between ADP-ribose and F-actin. The leaving group nicotinamide stays at its original position in the post-reaction state. Dissociation of TcART from ADP-ribosylated F-actin results in the reformation of the ionic bond between K185 and E265, closing the gate of the nucleotide-binding pocket (Fig. 3b).

To investigate if closing the gate would prevent TcART from rebinding to the already modified actin, we performed cosedimentation analyses with ADP-ribosylated F-actin. Indeed, we found that the binding affinity of TcART to the modified F-actin was much lower compared to unmodified F-actin (Fig. 3e, Supplementary Fig. 7a and j). Consistent with this, the E265S TcART variant lacking the K185-E265 gate retained its ability to bind to modified F-actin. Both findings are in line with the steric clashes expected for the binding of a closed apo TcART that would not occur in case of the E265S variant (Fig. 3d). Overall, our results demonstrate that TcART employs a gate mechanism to prevent rebinding of the toxin to the modified substrate. E265 plays therefore a crucial role not only in the ADP-ribosylation reaction and in NAD$^+$ binding, but also in increasing enzymatic turnover by preventing futile substrate encounters.

**Structure of the ADP-ribosylated F-actin**. The equilibrium between actin in G- and F-states is tightly regulated by numerous actin-binding proteins. In particular, thymosin β4 forms a complex with G-actin to form a reservoir of non-polymerizable actin that can be mobilized by competition with other monomer-binding proteins such a profilin. Previously, it was suggested that TcART-mediated ADP-ribosylation of T148 of actin sterically impairs the interaction between G-actin and thymosin β4 leading to aberrant, excess actin polymerization in cells[18]. However, our biochemical and structural data clearly show that TcART modifies actin filaments, but not G-actin, suggesting a different

mechanism of action. To decipher the structural basis of this mechanism, we performed a single-particle cryo-EM analysis of ADP-ribosylated F-actin (Supplementary Fig. 10, Supplementary Table 2).

We obtained a 3D reconstruction of the ADP-ribosylated actin filament at an average resolution of 3.5 Å, which allowed us to build an atomic model for F-actin including the ADP-ribose (Fig. 4a). The density corresponding to the ADP-ribose appears only at a lower threshold and is not as well defined as the rest of the map, indicating that the ADP-ribose moiety is flexible and that probably not all actin subunits are modified. Therefore, it is unlikely that the ADP-ribose modification itself is sufficient to stabilize actin-actin interactions to an extent that would lead to a toxic effect. Indeed, previously published experiments have demonstrated that the critical concentration of polymerization of ADP-ribosylated actin is almost identical to that of native actin[18,21]. However, when we compared our model of ADP-ribosylated F-actin with available structures of F-actin in complex with actin-binding proteins, we found that the ADP-ribose moiety occupies the same position on F-actin as the major actin-depolymerizing factor cofilin (Fig. 4a). Steric hindrance of the cofilin-F-actin interaction by the ADP-ribose could provide an alternative explanation for the increase in bulk amount of actin filaments in toxin-injected cells.

To test this mechanism, we employed TIRF-microscopy to study the influence of ADP-ribosylation on cofilin-induced actin depolymerization of individual actin filaments in vitro. Strikingly, we observed no binding of cofilin and no severing of ADP-ribosylated actin filaments under conditions where wild-type filaments were rapidly bound and severed (Fig. 4b, c). This confirmed that the modification impairs the formation of the cofilin-F-actin complex. Consistent with these results, it was previously demonstrated that ADP-ribosylated F-actin can also not be depolymerized by gelsolin or its fragments[21]. In conclusion, ADP-ribosylated F-actin is impaired in interacting with actin-severing and destabilizing proteins, resulting in an inhibition of actin turnover and the accumulation of excess actin filaments. Thus, our results uncover the mechanism of inhibition of actin turnover by a bacterial toxin.

## Discussion

Actin is a highly conserved protein that is involved in essential cellular processes such as cytokinesis, vesicle transport, migration, and phagocytosis[33]. As such, it is the target of a large number of ADP-ribosyltransferase toxins produced by a wide range of bacteria, including many human pathogens[34]. Interestingly, up to date, all but one of these enzymes modify G-actin at R177. In this manuscript, we focused on the single exception, the TcART enzyme that modifies F-actin at T148, and based on the structural snapshots of its activity, revealed its mechanism of action in atomic detail.

Following secretion by *P. luminescens*, the Tc toxin harbors its effector inside the TcB-TcC cocoon. Our NMR data indicate that TcHVR resides there in an unfolded state. Thus, the actual effector can be directly translocated through the narrow constriction site at the bottom of the cocoon as soon as TcB-TcC binds to TcA[12], and does not need be unfolded prior to translocation as in the case of anthrax toxin, for example[35–38]. This also explains why TcHVR was neither resolved in cryo-EM structures nor X-ray structures of TcB-TcC cocoons from *P. luminescens* and *Y. entomophaga*. Interestingly, the effector could also not be resolved in Rhs toxins from *Pseudomonas protegens* and *Photorhabdus laumondii*[39,40]. Similarly to TccC3, the N-terminal part of Rhs forms a cocoon that encapsulates the C-terminal effector region. Because the overall organization of the

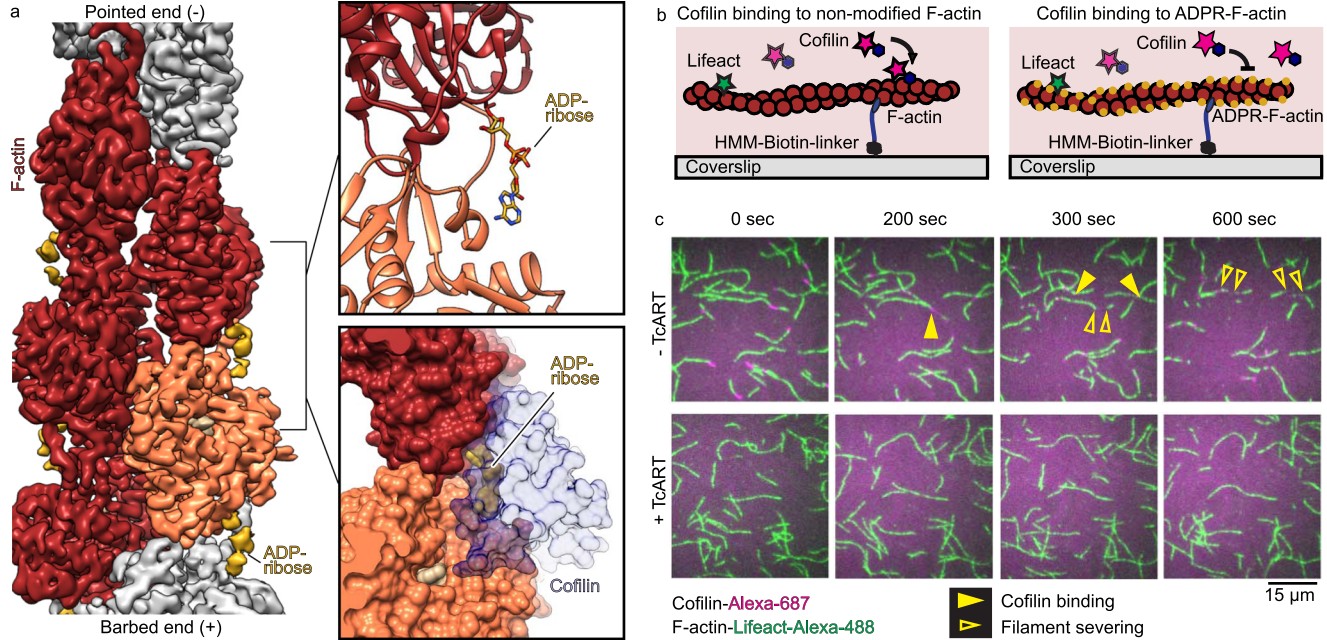

**Fig. 4 ADP-ribosylation of Thr-148 inhibits actin turnover by impairment of F-actin-cofilin interaction. a** Cryo-EM density, corresponding atomic model of ADP-ribosylated F-actin, and surface representation of the docking of cofilin onto ADP-ribosylated F-actin. Cryo-EM density for ADP-ribose (yellow) is shown at lower threshold than that of F-actin (orange and red). **b** Scheme of single-molecule TIRF microscopy assays. **c** Time-lapse images of non-modified or ADP-ribosylated actin filaments (green) in the presence of cofilin (magenta). Filled yellow arrows indicate the initial position of cofilin binding on the actin filament and empty yellow arrows indicate the moment of filament severing. The experiment was performed 3 times, and a representative result is shown.

cocoon is similar to that of Tc toxins, we assume that the effectors of these Rhs toxins are also unfolded, suggesting that this is a general property of cocoon-forming toxins.

Our cryo-EM structure of the TcART-F-actin complex revealed the extended interface between the toxin and its substrate (Supplementary Fig. 9). Interestingly, the position where TcART binds to F-actin partly overlaps with the position of other F-actin interacting proteins and peptides, such as myosin-V[41], ExoY toxin from *Pseudomonas aeruginosa*[42], and the Lifeact peptide[43,44]. In particular, all these proteins possess a hydrophobic residue that is homologous to Y183 of TcART (M515 in myosin-V, F374 in ExoY and F10 in Lifeact). These residues interact with a hydrophobic pocket that is formed at the interface between two actin subunits in F-actin and hence does not exist in G-actin. Therefore, recognition of this site seems to be a common property of many proteins that specifically bind to actin filaments.

Our model of the pre-reaction TcART-NAD+-F-actin state reveals that, similar to other bacterial ADP-ribosyltransferases, TcART orients NAD+ in a bent conformation. As a consequence, the pyridinium N-glycosidic bond is under strain, reducing the activation energy for the subsequent substitution reaction[45]. The weak potential of T148 as a nucleophile, the relatively large distance between the nucleophile and the electrophile (4.8 Å), and the presence of E265, which can stabilize an oxocarbenium ion of a possible intermediate substrate, suggests an $S_N1$ reaction. This type of reaction was proposed for the ADP-ribosylation reaction catalyzed by the iota toxin[30]. There, after cleavage of the glycosidic bond between nicotinamide and ribose, the first transition state oxocarbenium cation is stabilized by E380 and Y251. Then, a rotation releases the strain and brings the electrophile in close proximity of the nucleophile of the substrate. While ADP-ribosylation of F-actin by TcART can indeed follow a similar mechanism, we noted that TcART does not possess a homolog of Y251 and the charge of E265 is partly neutralized by K182, thus impairing its potential in stabilizing the oxocarbenium ion.

Therefore, as an equivocal alternative, we hypothesize that E265 only stabilizes the 2'-OH group of ribose in order to fix the anomeric center exactly in the right position for a $S_N2$-like reaction. Although the process may happen by itself because of the strained conformation of NAD+ inside the catalytic center and the leaving group nicotinamide being a weak base, an additional water may help to break the pyridinium N-glycosidic bond to bring the intermediate ion closer to T148 for a nucleophilic attack.

Numerous human pathogenic bacteria produce ADP-ribosyltransferases that modify R177 of G-actin. Although the modified actin cannot polymerize by itself, it binds to the barbed ends of existing actin filaments to prevent further elongation[27,46]. Since the pointed ends remain free, these capped filaments depolymerize. This mechanism of toxicity differs from that of TcART, which modifies T148 on the surface of the actin filament. While our data does not exclude that ADP-ribosylation affects the subtle equilibrium at the barbed end of F-actin, thus accelerating actin polymerization[18], we believe that the modification itself does not sufficiently stabilize actin-actin interactions in the middle of filaments. However, we clearly demonstrate that the addition of the bulky ADP-ribose group to this amino acid impedes binding of the actin-depolymerizing factor cofilin and other actin-binding proteins[21], thereby interfering with actin turnover, which leads an uncontrolled actin polymerization and ultimately to an accumulation of actin filaments in the intoxicated cells. Thus, unlike R177-modifying enzymes, ADP-ribosylation of T148 stabilizes actin filaments indirectly by hindering cofilin-mediated depolymerization. Nevertheless, both types of toxins effectively disrupt the regulation of the actin cytoskeleton with severe consequences for the host.

Our structural data allow us to describe the molecular mechanism underlying the threonine ADP-ribosylation of F-actin by a Tc toxin. Following secretion by *P. luminescens*, the Tc toxin harbors its effector inside the cocoon in an unfolded state (Fig. 5).

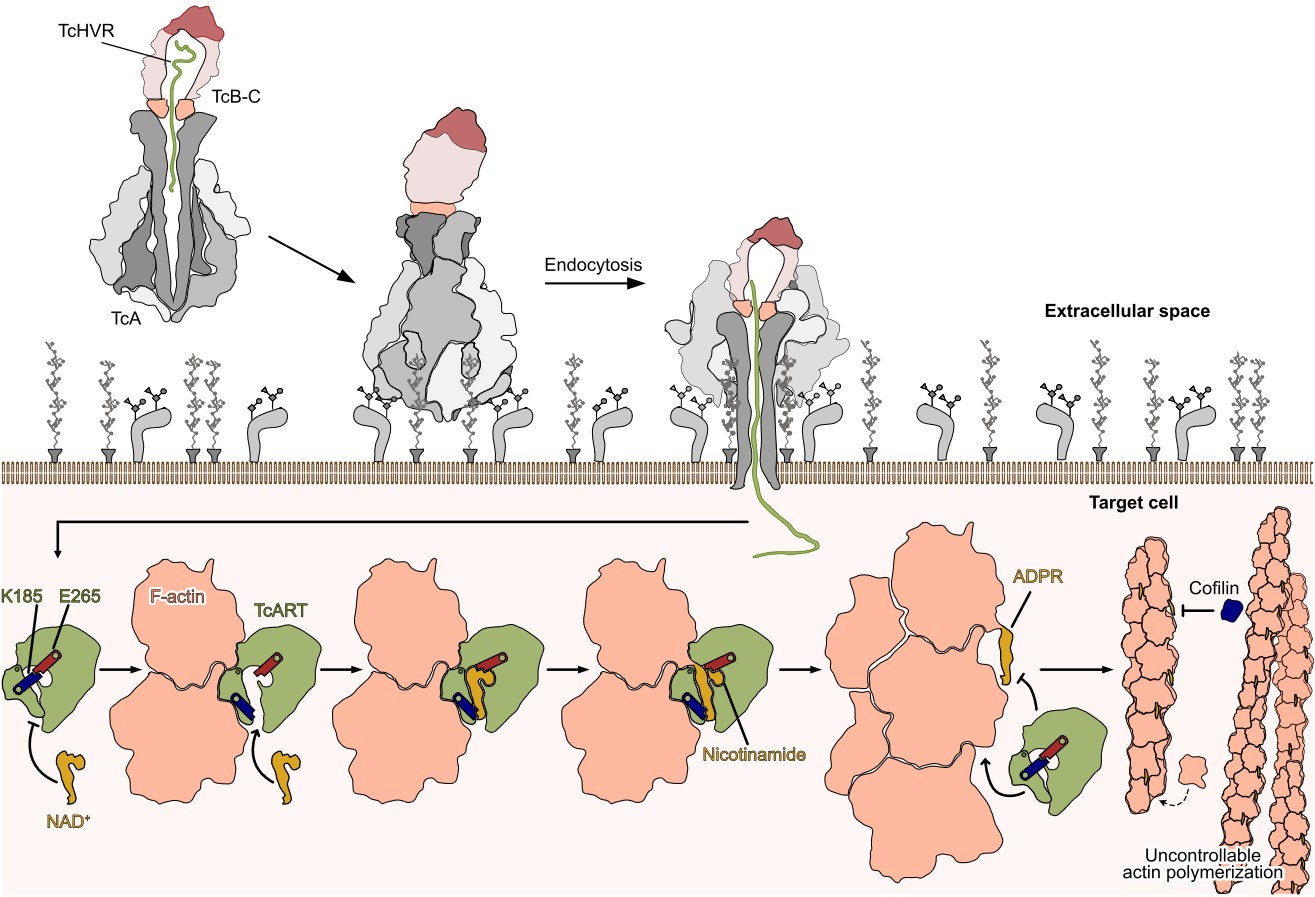

**Fig. 5 Mechanism of TcHVR-induced intoxication.** After binding of the toxin to its receptor(s) on the surface of the target cell, the Tc complex is endocytosed. Inside the late endosome, the Tc toxin undergoes the prepore-to-pore transition and the TcHVR is translocated into the host cell cytoplasm. There, TcHVR folds and interacts with F-actin via an induced-fit mechanism that facilitates $NAD^+$ access to the nucleotide-binding pocket. The following reaction results in the transfer of ADP-ribose to threonine-148 of F-actin. The modification impairs the interaction of F-actin with actin-depolymerizing factors, which inhibits actin turnover and eventually leads to cell death.

After glycan-dependent cell adhesion[13] and binding of the toxin to as-yet-unknown protein receptors on the surface of the host cell, the Tc complex is endocytosed. In the late endosome, the toxin undergoes a pH-triggered prepore-to-pore transition, entering the host membrane and the HVR domain is translocated into the cytoplasm of the host cell[15]. There, TcHVR folds in a spontaneous[14] or chaperone-assisted process[16] and interacts specifically with F-actin. Although $NAD^+$ can bind directly to the toxin, it is more likely that TcART first binds to F-actin by an induced-fit mechanism that opens the K185-E265 gate, facilitating binding of $NAD^+$ to the complex. By this process the nucleotide is placed in close proximity to the hydroxy group of T148 of actin, which performs a nucleophilic attack on the anomeric carbon of the N-ribose, resulting in ADP-ribose covalently bound to F-actin and nicotinamide as the leaving group. After ADP-ribosylation, the toxin dissociates from the filament and restores the salt bridge between K185 and E265, thereby obstructing the nucleotide-binding pocket. This mechanism prevents the enzyme from re-binding to the already modified F-actin. The ADP-ribosylation itself is relatively flexible and does not stabilize the filament. Instead, it impairs the interaction of F-actin with actin-depolymerizing factors, which inhibits actin turnover and ultimately leads to cell death.

## Methods

### Plasmids, bacteria and yeast strains, growth conditions.
The complete list of oligonucleotides, plasmids and strains can be found in the Supplementary Table 3.

*E. coli* were cultivated in LB medium supplemented with kanamycin or ampicillin. *S. cerevisiae*, transformed using the lithium-acetate method[47], were grown on synthetic defined medium (Yeast nitrogen base, Difco) containing galactose or glucose and supplemented if required with histidine, uracil, tryptophan, adenine, or leucine. Yeast viability upon toxin expression was analyzed by a drop test[48]. The analysis of protein expression in yeast was performed using the following method: yeast cells were grown overnight at 30 °C in liquid galactose–containing medium. Then, cells corresponding to 1 ml of $OD_{600}$ 1.0 were washed with 0.1 M NaOH, resuspended in 50 μl of 4-fold Laemmli sample buffer, and boiled for 5 minutes at 95 °C. 8 μl of the extracts were separated by SDS-PAGE, followed by Western blotting analysis, and incubation with anti-myc (dilution 1 to 10000, clone 9B11 #2276, Cell Signaling Technology), anti-mono-ADP-ribose binding reagent (dilution 1 to 5000, MABE1016, Merck) or anti-RPS9 serum (dilution 1 to 10000, polyclonal rabbit antibodies were a generous gift of Prof. S. Rospert), and visualized using secondary anti-mouse HRP (dilution 1 to 5000, #1706516, Bio-Rad) or anti-rabbit HRP-antibodies (dilution 1 to 5000, #1706515, Bio-Rad). Exchange of yeast actin alleles was performed as previously described[48]. In short, plasmids encoding yeast actin gene with point mutations were transformed into SC483. Then, the WT *ACT1* was removed by incubating yeast cells on the selective media supplemented with 5-fluoroorotic acid.

### Protein expression and purification.
Fusion proteins of TcHVR and maltose-binding protein (MBP) were expressed overnight at 22 °C in the presence of 0.1 mM IPTG in *E. coli* BL21-CodonPlus(DE3)-RIPL cells possessing the plasmids listed in Supplementary Table 3. Following the lysis in the buffer containing 20 mM Tris pH 8 and 500 mM NaCl, the soluble fraction was loaded onto Ni-IDA column, washed with the lysis buffer, and eluted with the same buffer supplemented with 250 mM imidazole. The eluates were dialyzed against the buffer with 20 mM Tris pH 8 and 150 mM NaCl, and stored in aliquots at −20 °C.

Isotope-labeled TcHVR and TcART in *E. coli* BL21-CodonPlus(DE3)-RIPL were expressed in 2x M9 minimal medium supplemented with 100 μg/mL ampicillin, 10 mL trace elements solution (1.25 g/L ETDA, 125 mg/L $FeSO_4$, 12.5 mg/L $ZnCl_2$, 2.5 mg/L $CuSO_4$) per L, 3 ug/L Thiamin-HCl, 3 μg/L Biotin, 4 g/L

glucose and 1 g/L $NH_4Cl$. For $^{15}N$-labeled protein, $^{15}N$-labeled $NH_4Cl$ (Sigma) was added. For $^{13}C$, $^{15}N$-labeled protein, $^{13}C$-labeled glucose (Cortecnet) and $^{15}N$-labeled $NH_4Cl$ was added. For $^{2}H$, $^{13}C$, $^{15}N$-labeled protein, $^{2}H$, $^{13}C$-labeled glucose (Cortecnet) and $^{15}N$-labeled $NH_4Cl$ was added, and the medium was prepared in $D_2O$. Expression cultures of 1 L were inoculated with an overnight starter culture (2x M9 medium) from a single colony and incubated at 37 °C until an $OD_{600}$ of ~0.6, after which the temperature was reduced to 25 °C and expression was induced with 0.2 mM IPTG. After 24 h of expression (30 h in case of expression in $D_2O$), the cells were pelleted and lysed in 25 mM Tris-HCl pH 8, 300 mM NaCl, 5% glycerol, 10 mM imidazole, 0.5 mM TCEP using a microfluidizer. The soluble bacterial extract was applied to the Ni-NTA column and washed with lysis buffer, followed by elution with a gradient from 10 to 250 mM imidazole in lysis buffer. The eluted protein was dialyzed against 20 mM Tris-HCl pH 8.0, 150 mM NaCl for 2 h and cleaved using hexahistidine-tagged Human Rhinovirus 3C protease (0.05 mg per mg of TcHVR or TcART) for 16 h at 4 °C. Subsequently, the protein solution was again applied to a Ni-NTA column and the protein of interest-containing flow-through was subjected to size exclusion chromatography on a Superdex 75 16/60 column in 20 mM Hepes-NaOH, 150 mM NaCl, 0.5 mM TCEP.

Unlabeled wild-type TcdB2-TccC3 was expressed in fusion with an N-terminal hexahistidine tag in *E. coli* BL21-CodonPlus(DE3)-RIPL in LB medium and purified essentially as described previously[12]. $^{15}N$-labeled TcdB2-TccC3 was expressed in the same medium as described for TcHVR, supplemented with 50 mg/L kanamycin and 30 μM IPTG. 3 L of medium were inoculated with a fresh transformant and incubated at 28 °C for 30 h, followed by 24 h at 22 °C. The cells were pelleted and disrupted in lysis buffer (20 mM Tris-HCl pH 8.0, 300 mM NaCl, 10% glycerol) using a microfluidizer. The soluble extract was applied to a Ni-NTA column and washed with the lysis buffer supplemented with 40 mM imidazole, followed by elution with a linear gradient from 40 mM to 250 mM imidazole in lysis buffer. The eluted protein was diluted with 20 mM Tris-HCl pH 8.0, 5% glycerol to a final NaCl concentration of 20 mM and loaded on a HiTrapQ column. After washing with 20 mM Tris-HCl pH 8.0, 20 mM NaCl, 5% glycerol, the protein was eluted with a linear gradient from 20 to 500 mM NaCl. Fractions containing TcdB2-TccC3 were loaded on a Superdex 200 10/300 column equilibrated in gel filtration buffer (20 mM Tris-HCl pH 8.0, 150 mM NaCl, 5% glycerol). The purified protein was then dialysed to 25 mM Hepes-NaOH pH 8.0, 200 mM NaCl, 25 % glycerol, flash-frozen in liquid nitrogen and stored at −80 °C.

Mammalian cytoplasmic beta-actin T148N with an additional mutation Cys272Ala to avoid its oxidation was purified as described previously[43]. In brief, actin was expressed in insect cells BTI-Tnao38 (Boyce Thompson Institute for Plant Research, Inc.) as a C-terminal fusion with thymosin β4 and a His-tag. After baculovirus-mediated expression, cells were resuspended and lysed in the buffer containing 10 mM Tris pH 8, 50 mM KCl, 5 mM $CaCl_2$, 1 mM ATP, 0.5 mM TCEP and cOmplete protease inhibitor (Sigma). The supernatant after centrifugation was loaded on the HisTrap FF column (ThermoFisher Scientific). After the washing step with the resuspension/lysis buffer, actin was eluted with a gradient of imidazole. After overnight dialysis in G-buffer (5 mM Tris pH 8, 2 mM $CaCl_2$, 0.5 mM ATP and 0.5 mM TCEP), actin was mixed with chymotrypsin and incubated for 20 minutes at 25 °C to cleave off thymosin β4 and the following His-tag. After stopping the reaction by addition of PMSF to the final concentration of 0.2 mM, the mixture was applied again onto the HisTrap FF column. The actin-containing flow-through was collected and polymerized overnight by the addition of KCl and $MgCl_2$ (100 mM and 2 mM final concentration, respectively). The next day actin was spun down at $210,000 \times g$ for 1 h. The F-actin pellet was resuspended in G-buffer and dialyzed for at least 3 days against G-buffer. Finally, the protein was spun down at $210,000 \times g$ for 1 h, concentrated on 10 kDa cutoff Amicon columns, flash-frozen in liquid nitrogen and stored in small aliquots at −80 °C.

We purified rabbit skeletal muscle alpha-actin as described previously[42]. In short, muscle acetone powder (a generous gift of W. Linke and A. Unger, Ruhr-Universität Bochum, Germany) was first resuspended in G-buffer (5 mM Tris-HCl, pH 7.5, 1 mM DTT, 0.2 mM $CaCl_2$, 0.5 mM ATP). Then, the solution was spun down for 30 minutes at $100.000 \times g$ to remove solid particles and debris. The G-actin-containing supernatant was mixed with $MgCl_2$ and KCl (2 mM and 100 mM final concentration, respectively) to induce actin polymerization. After 1 h of incubation at room temperature, followed by the addition of KCl to the final concentration of 800 mM to release actin-binding proteins, F-actin was centrifuged for 2 hours at $100.000 \times g$. The F-actin pellet was then dialyzed against G-buffer for 2 days to depolymerize actin, and actin was polymerized and depolymerized once again. The final G-actin was flash-frozen in liquid nitrogen and stored in small aliquots at −80 °C.

*Clostridium perfringens* Iota-A toxin was expressed and purified as described previously with minor modifications[48]. In short, *E. coli* were transformed with plasmid 1330 and grown on a shaker at 37 °C until $OD_{600}$ reached 1. After overnight expression at 28 °C in the presence of 1 mM IPTG, the cells were collected and suspended in the lysis buffer (20 mM Tris-HCl pH 8, 0.5 M NaCl). After lysis by sonication, the bacterial extract was loaded onto Ni-IDA resin. The resin was then washed with the lysis buffer, and the protein of interest was eluted with the lysis buffer supplemented with 250 mM imidazole and dialyzed overnight against 20 mM Tris-HCl pH 8 with 150 mM NaCl.

Human cofilin-1 was purified as described previously[43]. In brief, *E. coli* cells were transformed with plasmid 1855. An overnight culture was then diluted 100

times in fresh LB media and grown at 37 °C. When $OD_{600}$ reached ~1, the cells were cooled down to 30 °C, and cofilin expression was induced by adding IPTG (final concentration 0.5 mM). After 4 h of expression, the cells were harvested, resuspended in the lysis buffer (10 mM Tris pH 7.8, 1 mM EDTA, 1 mM PMSF, and 1 mM DTT), and disrupted using a fluidizer. After lysis, the extract was dialyzed overnight in a buffer containing 10 mM Tris pH 7.8, 50 mM NaCl, 0.2 mM EDTA, and 2 mM DTT, and cleared by centrifugation. Then, the lysate was applied onto DEAE resin and washed with the dialysis buffer. Cofilin-containing fractions of the flow-through were collected and dialyzed against low-pH buffer, containing 10 mM PIPES pH 6.5, 15 mM NaCl, 2 mM DTT, and 0.2 mM EDTA. The next day, the protein was loaded onto Mono S column and eluted by a linear gradient of 15 mM to 1 M NaCl in the low-pH buffer. Cofilin-containing fractions were concentrated, flash-frozen in liquid nitrogen and stored at −80 °C.

**ADP-ribosylation assays**. 1 μg of actin (in G-, or F- state, with latrunculin or phalloidin, if indicated) in the assay buffer (1 mM $NAD^+$, 20 mM Tris pH 8, 150 mM NaCl, and 1 mM $MgCl_2$) was added to the indicated amount of TcHVR or Iota-A in the total volume of 10 μl. After 10 minutes of incubation at 37 °C, the reaction was stopped by adding Laemmli sample buffer (62.5 mM Tris-HCl pH 8.0, 25 mM DTT, 1.5% SDS, 10% glycerol, and 0.1 mg/ml bromphenol blue) and heating the sample at 95 °C for 5 minutes. Components of the mixture were separated by SDS-PAGE, blotted onto a polyvinylidene difluoride membrane using a Trans-Blot Turbo Transfer System, and visualized using a combination of anti-mono-ADP-ribose binding reagent (dilution 1 to 5000, MABE1016 Merck) and anti-rabbit-HRP antibody (dilution 1 to 5000, #1706515, Bio-Rad).

**Cosedimentation assays**. An aliquot of freshly thawed G-actin was centrifuged at $150,000 \times g$ using a TLA-55 rotor for 20 min at 4 °C to remove possible aggregates. Then, actin was polymerized by incubation in F-buffer (120 mM KCl, 20 mM Tris pH 8, 2 mM $MgCl_2$, 1 mM DTT, and 1 mM ATP) for 2 h at room temperature. To stabilize filaments, phalloidin was added in 1.5 excess over actin after polymerization.

Cosedimentation assays were performed in 20-μl volumes by first incubating F-actin with the specified amount of proteins (in the presence of 1 mM of $NAD^+$, if indicated) for 5 min at room temperature, then centrifuging at $120,000 \times g$ using the TLA120.1 rotor for 20 min at 4 °C. After centrifugation, aliquots of the supernatant and pellet fractions were separated by SDS-PAGE and analyzed by densitometry using Image Lab software version 5.2.1 (Bio-Rad) and Prism version 9 (GraphPad Software).

**TIRF microscopy**. F-actin was prepared as for cosedimentation assays. Then, the filaments were spun down at $150,000 \times g$ using a TLA-55 rotor for 20 minutes at 4 °C and resuspended in TIRF buffer (20 mM Hepes pH 7.0, 100 mM KCl, 1.5 mM $MgCl_2$, 1 mM EGTA, 20 mM β-mercaptoethanol, 0.1 mg/ml β-casein, 0.2% methylcellulose (cP400, M0262, Sigma-Aldrich), 1 mM ATP and 2 mM Trolox). Individual pre-polymerized and aged actin filaments (either WT or ADP-ribosylated) were bound to Biotin-PEG-functionalized glass coverslips[49] via HMM-linker molecules and were visualized by trace amounts (10 nM) of Lifeact-Alexa488[50] in the presence of 250 nM Alexa647 labeled cofilin 1. TIRF imaging was performed on a customized Nikon TIRF Ti2 microscope equipped with dual camera EM CCD Andor iXon system (Cairn) controlled by Nikon Elements AR 4.50 software. Dual-colour imaging was performed using an Apo TIRF 60x oil DIC N2 objective and a custom multilaser launch system (AcalBFi LC) at 488 nm and 640 nm. Data were analyzed in Fiji version 1.53C.

**NMR spectroscopy**. Solid-state NMR spectroscopy measurements of the $^{1}H$, $^{15}N$-labeled TcB-TcC complex (with and without TcHVR) were conducted at 100 kHz magic-angle spinning (MAS) and 20 °C using a Bruker 900 MHz spectrometer equipped with a 0.7 mm probe in a narrow bore 21.14 Tesla magnet. Temperature calibration was performed on a water/DSS sample by monitoring the water peak in reference to DSS. The rotor was filled and the 2D hNH spectra at 100 kHz were recorded as described previously[51]. Further parameters concerning acquisition and processing (TopSpin 3.5pl6 for the solution and TopSpin 4.1.0 for solid-state NMR) of all acquired spectra can be found in Supplementary Table 4. Backbone assignments were achieved by evaluating 3D HNCO, HNCACB, HNCOCACB spectra[52] of $^{2}H$, $^{13}C$, $^{15}N$-labeled protein in 90% $H_2O$, 10% $D_2O$ buffer (20 mM Hepes, 150 mM NaCl, pH 7.8) measured at 280 K. Side chain assignments were obtained from combined evaluation of 3D HNCA and CCH-TOCSY experiments[52] recorded on $^{13}C$, $^{15}N$-labeled sample in the same buffer and under identical conditions. Certain assignments (aromatic carbon and proton signals, arginine side chain signals) were based on the evaluation of two 3D CHH-and a NHH-NOESY-type spectra[52] of the same sample, all with a mixing time of 80 ms. The two CHH-NOESY spectra focused either on aromatic or on aliphatic resonances. Very slow exchanging amide protons and chemical shifts of Hα close to the water were identified after dialysis of the $^{13}C$, $^{15}N$-sample into a 100% $D_2O$ buffer, containing 20 mM d11-Tris instead of Hepes. Peak lists containing structural restraints were obtained from 2D NOESY (40 ms mixing), the 3D NOESY spectra above, and a CHH-NOESY (40 ms mixing) spectrum in $D_2O$ buffer.

$^{15}$N relaxation rates ($R_1$, $R_2$) were measured as pseudo-3D experiments[53] at 600 MHz $^1$H Larmor frequency, 280 K and with the buffer described above. Individual experiments with $R_1$ (0.01, 0.05, 0.1, 0.2, 0.4, 0.6, 0.9 and 1.5 s) or $R_2$ (0.015, 0.035, 0.04, 0.06, 0.08, 0.13, 0.17 and 0.22 s) delays were recorded in a scrambled manner and the FID acquired in an interlaced fashion. Peak heights from resulting $^1$H-$^{15}$N HSQC spectra were extracted using CcpNMR version 2.4.2[54], normalized and fit to a mono-exponential decay ($f(x) = a \cdot e^{-b \cdot x}$) using the $curve\_fit$ function of the python module $SciPy$ for $R_1$ and $R_2$. Standard deviations were calculated from the diagonal of the returned covariance matrix ($pcov$) of the $curve\_fit$ function.

Two-dimensional $^1$H-$^{15}$N correlation spectra (HSQC) without and with a threefold NAD$^+$ ligand concentration were recorded to determine chemical shift perturbations. A weighted shift distance $d$ as shown in Fig. 1d and Supplementary Fig. 5 was calculated as $d = \sqrt{\delta_H^2 + (0.14 \cdot \delta_N)^2}$ [55]. All assignments were conducted with CcpNMR version 2.4.2[54].

**NMR structure calculations.** Structure calculation was performed with an iterative NOE assignment procedure using the software ARIA version 2.3.2[56] coupled to CNS version 1.21[57]. Inter-residue cross-peaks from NOESY-type spectra were submitted to ARIA for several cycles of automated NOE assignment and structure calculation. To ensure a better representation of the conformational space allowed from the NOE-derived distance restraints and to prevent over-convergence to a possibly artefactual single state conformation, we implemented a $consensus$ procedure into ARIA, following reports by[58]. A script to perform consensus calculations with ARIA is available at http://aria.pasteur.fr. In short, 20 ARIA runs were first performed independently using the same input data but with different random number seeds, resulting in different random starting conformations and initial velocities for the molecular dynamics simulated annealing protocol. Then, cross-peaks that remained active (i.e., for which at least one assignment possibility was kept) at the end of 12 out of the 20 ARIA runs were collected. For each of the active cross-peaks, the assignment possibilities from each individual ARIA run were combined to yield a new list of consensus (ambiguous) distance restraints. Finally, a new ARIA run is performed with a single iteration and using the consensus distance restraints as input to produce the final consensus structure ensemble. For all ARIA runs, the NOE data were supplemented with backbone dihedral angle restraints derived from TALOS+[59] predictions based on Hα, H, N, Cα, Cβ secondary-chemical shifts and also hydrogen bond restraints based on H-D exchange data and secondary structure pattern. In the individual ARIA runs, nine iterations were performed with an adaptive tolerance procedure to discard unsatisfied distance restraints[60] and restraint combination (4->4)[61] was employed for the first 4 iterations. At each iteration, 50 conformers were calculated (except for the last one where 100 conformers were generated). The consensus calculation was run twice, and in each case a single iteration was performed generating 200 conformations of which the 15 lowest-energy ones were refined in a shell of water molecules[62]. Of both water refinements, the 5 lowest energy structures were selected to represent the final coordinates (Supplementary Fig. 6d). A log-harmonic energy potential with the optimal weighting of distance restraints was always applied during the simulated annealing[63]. To improve convergence, the number of molecular dynamics steps at the two cooling stages of the simulated annealing runs was increased to 40,000 for the individual runs and to 100,000 for the consensus runs. Violated restraints were analyzed and assignments manually corrected. Statistics concerning structures and restraints are reported in Supplementary Table 5, together with excerpts of the PSVS report.

**Cryo-EM of the TcART-F-actin complex.** The freshly thawed mammalian beta-actin was spun down using TLA-120 rotor for 20 minutes at 120,000 × g at 4 °C, and the supernatant with G-actin was collected. Then, the protein was polymerized overnight at 4 °C by incubation in F-buffer (120 mM KCl, 20 mM Tris pH 8, 2 mM MgCl$_2$, 1 mM DTT and 1 mM ATP). The next day, a mixture of 15 μM of F-actin, 25 μM MBP-TcART E265S, 1 mM NAD$^+$ was incubated for 60 min at 30 °C. Shortly before plunging, the mixture was diluted 4 times with F-buffer supplemented with 0.02% (w/v) of Tween-20 to improve ice quality. Plunging was performed using the Vitrobot Mark IV system (Thermo Fisher Scientific) at 13 °C and 100% humidity. 3 μl of the sample were applied onto a freshly glow-discharged copper R2/1 300 mesh grid (Quantifoil), blotted for 8 s on both sides with blotting force -15 and plunge-frozen in liquid ethane.

The dataset was collected using a Krios Titan transmission electron microscope (Thermo Fisher Scientific) equipped with an XFEG at 300 kV and CS-corrector using the automated data-collection software EPU version 2.7 (Thermo Fisher Scientific). 4 images per hole with defocus range of -0.5 – -2.5 μm were collected with K3 detector (Gatan) operated in super-resolution mode. Image stacks with 60 frames were collected with the total exposure time of 4 sec and total dose of 84.9 e$^-$/Å$^2$. 12284 images were acquired and 9840 of them were used for processing. Motion correction and CTF estimation have been performed in CTFFIND4[64] version 4.1.13 and MotionCorr2[65] version 1.3 during image acquisition using TranSPHIRE[66] version 1.4.28. The filament picking was performed using crYOLO version 1.8[67]. On the next step, 2.22 million helical segments were classified in 2D using ISAC[68] (Sphire package version 1.3) to remove erroneous picks. The remaining 2.03 million particles were used in the first 3D refinement in Meridien[69] (Sphire package version 1.3) with 25Å low-pass filtered F-actin map as initial model and with a spherical mask with a

diameter of 280 Å. Then, this refinement was repeated with a wide mask in the shape of the TcART-F-actin complex that was created from an intermediate iteration of the previous refinement. As due to the low occupancy of TcART its density was worse than that on F-actin, we performed a round of alignment-free 3D classification in Relion 3[70] with a mask covering one TcART and two actin subunits. Indeed, after removal of 1.59 million particles and following 3D refinement we obtained a reconstruction where TcART appeared at the same threshold as F-actin. Further particle polishing in Relion 3[70] and the final 3D refinement improved the overall resolution of the EM-density that was further used in the modelling.

To build a model of TcART-F-actin complex, we performed a fitting of NMR-model of TcART, ADP-ribose and nicotinamide that were modelled with eLBOW[71] (PHENIX package version 1.17) and F-actin (PDB 5ONV[72]) into the EM-density, followed by refinement in ISOLDE[73] version 1.0B4 and Phenix[74] version 1.17. Figures were prepared in UCSF Chimera version 1.14.

**Cryo-EM of ADP-ribosylated-F-actin.** The freshly thawed mammalian beta actin was spun down using TLA-120 rotor for 20 minutes at 120,000 × g at 4 °C, and the supernatant with G-actin was collected. Then, the protein at 20 μM was polymerized for 2 hours by incubation in F-buffer (120 mM KCl, 20 mM Tris pH 8, 2 mM MgCl$_2$, 1 mM DTT and 1 mM ATP) in the presence of 1 mM NAD$^+$ and 0.2 μM of TcART. Shortly before plunging, F-actin was diluted to 4 μM with F-buffer supplemented with 0.02% (w/v) of Tween-20 to improve ice quality. Plunging was performed using the Vitrobot Mark IV system (Thermo Fisher Scientific) at 13 °C and 100% humidity. 3 μl of the sample were applied onto a freshly glow-discharged copper R2/1 300 mesh grid (Quantifoil), blotted for 8 s on both sides with blotting force -15 and plunge-frozen in liquid ethane.

The dataset was collected using a Krios Titan transmission electron microscope (Thermo Fisher Scientific) equipped with an XFEG at 300 kV and CS-corrector using the automated data-collection software EPU version 2.8 (Thermo Fisher Scientific). 4 images per hole with defocus range of -0.5 – -2.5 μm were collected with K3 detector (Gatan) operated in super-resolution mode. Image stacks with 60 frames were collected with the total exposure time of 4 sec and total dose of 82.3 e$^-$/Å$^2$. 5342 images were acquired and 2758 of them were used for processing. Motion correction and CTF estimation have been performed in CTFFIND4[64] version 4.1.13 and MotionCorr2[65] version 1.3 during image acquisition using TranSPHIRE[66] version 1.5.13. The filament picking was performed using crYOLO[67] version 1.8. On the next step, 2.51 million helical segments were classified in 2D using ISAC[68] (Sphire package version 1.4) to remove erroneous picks. The remaining 2.35 million particles were used in the first 3D refinement in Meridien[69] (Sphire package version 1.4) with 25 Å low-pass filtered F-actin map as initial model and with a wide mask in the shape of F-actin. Then, this refinement was repeated with a tighter mask in the shape of the ADPR-F-actin. Further particle polishing in Relion[70] version 3.1 and the final 3D refinement improved the overall resolution of the EM-density that was further used in the modelling.

To build a model of ADPR-F-actin complex, we first added ADP-ribose to F-actin structure (PDB 5ONV), fit it into the EM-density in ISOLDE[73] version 1.0B4, and performed the final refinement and Phenix[74] version 1.17. Figures were prepared in UCSF Chimera version 1.14.

**Docking of NAD$^+$ to cryo-EM and NMR structures.** Molecular docking was carried out using Glide[75] included in the Maestro 12v7 software package (https://www.schrodinger.com/maestro). Glide uses a series of hierarchical filters to search for possible ligand positions in receptor binding sites. The $receptor$ $grid$ for the binding site of the TcART-F-actin complex was set up using default parameters and by centering the grid on extra cryo-EM electron density in the NBP. Flexible docking of NAD$^+$ was carried out with $XP$ (extra precision) $settings$ utilizing standard $core$ $pattern$ $comparison$ with the ADP-ribose position in the post-reaction state as reference, applying a tolerance (RMSD) of 2.5Å. In total 5 very similar poses were obtained for NAD$^+$ and the best pose was chosen according to the docking score and fit with the NMR chemical shift experiments.

**Reporting summary.** Further information on research design is available in the Nature Research Reporting Summary linked to this article.

## Data availability

The coordinates for the cryo-EM structures of the TcART-F-actin complex and ADPR-F-actin have been deposited in the Electron Microscopy Data Bank under accession numbers EMD-14532 and 14533. The corresponding molecular models for TcHVR, ADPR-F-actin, and the TcART-F-actin complex have been deposited at the wwPDB with accession codes PDB 7ZBQ, 7Z7H and 7Z7I. The NMR datasets used in this study are available in the BMRB under accession codes 34717 (Assignment data of protonated TcART), 51438 (Assignment data of deuterated TcART) and 51478 (Relaxation data of TcART). The raw data generated during the current study are available from the corresponding authors on request. Source data are provided with this paper. Uncropped gels and Western blots can be found in Supplementary Fig. 11. We used the following previously published structures: 1GIQ, 5ZJ5, 4Z9D, 4TLV, 1PTO, 1WFX, 6E3A, 6RO0, 5ONV, 3B8H, 5BWM, 4H03. Source data are provided with this paper.

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

## Acknowledgements
We thank O. Hofnagel and D. Prumbaum for assistance with data collection and maintaining the EM facility, F. Merino for help in building atomic models for the cryo-EM structures, S. Bergbrede for the excellent technical assistance, D. Friedrich and P. Schmieder for help in NMR measurements, W. Linke and A. Unger for providing us with muscle acetone powder, S. Rospert for providing us with anti-RPS9 serum, N. Duclert-Savatier for technical help on consensus calculations, and S. Pospich, Y. Belyi, R.S. Goody for fruitful discussions. This work has been funded by the Max Planck Society (S.R.). A.B. was supported by an EMBO long-term fellowship and a stipend of the Humboldt foundation. H.O. acknowledges funding by the German Research Foundation (DFG), grant Os 106/17-1.

## Author contributions
S.R. and H.O. designed and supervised the project. A.B. prepared cryo-EM specimens, collected and analyzed EM data. F.L. collected and analyzed NMR data. A.B. and F.L. built the atomic models from cryo-EM and NMR data, respectively. B.B. modified ARIA/CNS to enable the generation of consensus constraint lists and consensus structure calculations. J.P. performed docking experiments. A.B. performed affinity and activity assays. D.R. produced proteins for NMR studies. J.F. performed and analyzed TIRF-microscopy with help of P.B. A.B., F.L., and J.F. prepared figures. A.B., F.L., H.O. and S.R. wrote the manuscript with input from all other co-authors.

## Funding

## Competing interests
The authors declare no competing interests.
