## [Peer Review File · Nature Communications]

REVIEWERS' COMMENTS

Reviewer #1 (Remarks to the Author):

Belyy et al. present a work that unravels the mechanism of the bacterial toxin Tc. Tc is injected into cells in an unfolded form and then folds and ribosylates actin to result in uncontrolled actin polymerization. The authors combine multiple structural biology methods including solution and solid-state NMR spectroscopy and cryo-electron microscopy to resolve this process mechanistically and at the atomic level.

This is a beautiful work that readily deserves publication in Nat. Comm. The work has highest quality, is clearly documented and the figures are very nice. The paper is well written and interesting. The authors are applauded for comprehensively unravelling the entire story including several structure determinations, rather than chopping the work into smaller publishable units. I highly recommend publication after addressing the following point.

The relaxation data in Fig. S4 uses an undefined convention for reporting the heteronuclear NOE. Usually, the $^{15}\text{N}, ^1\text{H}$ het NOE has a maximal possible value between 0.8 and 1, depending on field strength and molecular size. Values above 1 are not possible. Perhaps the authors are plotting here the enhancement? Or have a calibration error of some kind, please doublecheck and convert to standard convention. Furthermore, the measured relaxation rate constants R_1 and R_2 for the rigid part of the protein ($\sim 0.62 / 28 \text{ s}^{-1}$) correspond to a rotational correlation time of 21 ns for isotropic molecular tumbling. This is around twice as much as one would expect for this molecular size. Is the protein dimeric or can the authors somehow else rationalize the values, e.g. by anisotropic tumbling (or is there also a calibration problem in the data)? This seeming discrepancy should be resolved and commented on in the manuscript. The field strength (600 MHz) should be given in the Figure caption.

Reviewer #2 (Remarks to the Author):

This manuscript by Oschkinat, Raunser and colleagues provides a detailed biochemical and structural analysis of how a unique class of Tc toxins ADP-ribosylates F-actin and impairs its dynamics. By capturing a series of high quality structural snapshots of TcART alone and in complex with F-actin using both NMR and cryo-EM, the study presents unprecedented mechanistic insight into TcART's substrate specificity and mechanism. Using structural analysis and TIRF experiments, the authors

further demonstrate that ADP-ribosylated F-actin is resistant to severing by cofilin due to a steric clash between the ADP-ribose and cofilin. This provides a mechanistic explanation for how F-actin turnover is perturbed by the toxic modification in cells.

Overall, the experiments are well-designed and the results are clearly described. The study significantly advances our understanding of a unique type of F-actin modification by a bacterial toxin, which is likely of general interest to both the actin cytoskeleton and bacterial pathogenesis fields. Below are a few specific comments and minor points which I believe should be addressed through textual revisions prior to acceptance for publication:

Comments:

1 Page 4, paragraph 2: The authors discuss the presence and importance of two hydrogen bonds between Y112 and A195. I couldn't spot them anywhere in the figures. Please show them to support the argument.

2 Page 8, last paragraph: The authors claimed "D263 plays a central part in the enzymatic reaction and can't be replaced by other residues". Fig. S7g clearly showed the enzymatic activity of D263N is comparable to that of WT, which contradicts the authors' claim. Please tone down the statement.

3 Page 9, paragraph 2: The statement "suggesting that F-actin does not prefer TcART-NAD⁺ over ligand-free TcART" contradicts Fig2b showing a clear affinity difference for E265S with/without NAD⁺ and, to a lesser extent, for WT. The presence of NAD⁺ greatly increased the affinity of E265S to F-actin. This leads to a significant concern about the proposed mechanism in which the nucleotide-free state TcART interacts with F-actin first. Either NAD or F-actin could initiate binding to TcART and trigger an induced-fit mechanism. The current data is not sufficient to support one over the other. The authors need to either provide more evidence or tone this down a bit.

4 Page 12, paragraph 3: The authors postulated "it is unlikely that the ADP-ribose modification itself is sufficient to stabilize actin-actin interactions to an extent that would lead to a toxic effect." The clear density of ADP-ribose engages two adjacent actin subunits near the D-loop, which is a hotspot of F-actin polymerization regulation. This could support an alternative explanation in which ADP ribosylation perturbs actin dynamics by accelerating actin polymerization at the barbed end. Mannherz and colleagues (Lang, A.E. et al, 2017) demonstrated rapid actin polymerization in the presence of Tccc3 modified actin. What is your comment on this point? In their study, Tccc3

ribosylates both G-actin and F-actin, which differs from this manuscript. The authors should explain the discrepancy in the Discussion section.

5 In Fig S7e, E265S dramatically reduces ribosylation activity. I am confused why in the structure T148 is heavily ribosylated. What is equally confusing is why E265S mutant didn't dissociate from F-actin after ribosylation in the presence of excessive NAD⁺ in solution. A short explanation is needed.

6 Based on co-sedimentation assay, a significant portion of (>20%) wildtype TcART co-sediments with 16uM of actin. Are there any density for WT TcART during data processing? The authors didn't indicate the concentration of actin mixed with wildtype TcART in the Methods. Presumably it is the same as that with TcART E265S.

7 It is surprising that there is no binding of cofilin to ADP-ribosylated F-actin in the TIRF assay. I assume the filament is not fully ribosylated. Does that suggest ADP-ribosylation on T148 induces long range allosteric effect to the sites where T148 is not ribosylated? Are there structural differences between unmodified and ADP ribosylated F-actin? Lifeact interferes with cofilin severing. What is the percentage of Lifeact-F-actin used for the assay? Please add more details in the Methods.

Minor points:

1. Page 7, paragraph 2: add a space between (G-) and actin.
2. Page 9, paragraph 2: please correct "(Fig 3a, b, 5)".
3. In the Methods, please add the purification of actin T148N.
4. Figure S8a and S10a: please add length parameter to the scale bar.

Reviewer #3 (Remarks to the Author):

Prior studies show that the TccC3 hypervariable region ADP ribosylates actin residue T148, and that this leads to aberrant actin polymerization. Belyy et al present new data showing the structure of TccC3 hypervariable region, that the preferred substrate for the bacterial ADPr toxin TccC3 is F-actin, and that this modification prevents association and activity of depolymerizing factor cofilin.

The authors provide convincing evidence that the TccC3 hypervariable region is initially unfolded and positioned randomly inside the TcB-TcC cocoon using MAS NMR. They provide a rationale for their approach to determine the structure of TccC3 HVR, explaining why crystallography was not possible, and using NMR. They determine that the unstructured N-terminal region was not needed for toxicity, and propose it may serve as a linker to the TcB-TcC cocoon. Using informed mutagenesis, they were able to solve the structure of TccC3 HVR bound to F-actin by Cryo-EM, and test their results with mutagenesis of other key residues, showing it to differ from other RSE bacterial ARTs. Within this structure were densities for ADP ribose and nicotinamide, permitting them to investigate the most probable mechanism of ADP ribosylation. The authors also investigate how this modification leads to aberrant polymerization, showing it hinders association for cofilin using a previously published TIRF assay to monitor labeled actin filament dynamics with labeled cofilin.

The paper is well-written and easy to read. Inclusion of experiments in the text that did not yield results provide a clear rationale for the approaches the authors take. Data are clearly presented and the movie figure is a helpful addition. Overall this is an interesting addition to the Tc toxin and bacterial ART fields.

We thank the reviewers for their positive and constructive feedback, which aided us to further improve the manuscript. Below we include our detailed response to each point raised.

Reviewer #1 (Remarks to the Author):

Belyy et al present a work that unravels the mechanism of the bacterial toxin Tc. Tc is injected into cells in an unfolded form and then folds and ribosylates actin to result in uncontrolled actin polymerization. The authors combine multiple structural biology methods including solution and solid-state NMR spectroscopy and cryo-electron microscopy to resolve this process mechanistically and at the atomic level.

This is a beautiful work that readily deserves publication in Nat. Comm. The work has highest quality, is clearly documented and the figures are very nice. The paper is well written and interesting. The authors are applauded for comprehensively unravelling the entire story including several structure determinations, rather than chopping the work into smaller publishable units. I highly recommend publication after addressing the following point.

We thank the reviewer for the very positive feedback on our manuscript.

The relaxation data in Fig. S4 uses an undefined convention for reporting the heteronuclear NOE. Usually, the $^{15}\text{N}, ^1\text{H}$ het NOE has a maximal possible value between 0.8 and 1, depending on field strength and molecular size. Values above 1 are not possible. Perhaps the authors are plotting here the enhancement? Or have a calibration error of some kind, please doublecheck and convert to standard convention.

We are sorry, this really escaped the attention of a number of NMR specialists. We thoroughly investigated the raw data and found an issue with the parameters employed for the reference measurement. Thank you so much for pointing this out. In response, we have removed the heteroNOE data from Supplementary Fig. S4 since the message presented in the paper is already sufficiently substantiated by the R1 and R2 data presented in the same Figure.

Furthermore, the measured relaxation rate constants R1 and R2 for the rigid part of the protein ($\sim 0.62 / 28 \text{ s}^{-1}$) correspond to a rotational correlation time of 21 ns for isotropic molecular tumbling. This is around twice as much as one would expect for this molecular size. Is the protein dimeric or can the authors somehow else rationalize the values, e.g. by anisotropic tumbling (or is there also a calibration problem in the data)? This seeming discrepancy should be resolved and commented on in the manuscript. The field strength (600 MHz) should be given in the Figure caption.

The isotropic molecular tumbling depends on a number of local and global factors characteristic for each sample and the measurement conditions. We have calculated the rotational correlation time τ_c and relaxation rates for our NMR structure using hydromr (<https://doi.org/10.1006/jmre.2000.2170>), assuming the viscosity of water for 280 K ($\eta=1.4 \text{ mPa}\cdot\text{s}$), 293 K ($\eta=1 \text{ mPa}\cdot\text{s}$) and 300 K ($\eta=0.085 \text{ mPa}\cdot\text{s}$). The predicted τ_c of 21.4 ns and average relaxation rates ($\overline{R1} = 0.655 \text{ s}^{-1}$; $\overline{R2} = 29.98 \text{ s}^{-1}$) were in accordance with our experimental values attained at 280K, 600 MHz Larmor frequency, and buffer conditions (150 mM NaCl, 20 mM Hepes).

Reviewer #2 (Remarks to the Author):

This manuscript by Oschkinat, Raunser and colleagues provides a detailed biochemical and structural analysis of how a unique class of Tc toxins ADP-ribosylates F-actin and impairs its dynamics. By capturing a series of high quality structural snapshots of TcART alone and in complex with F-actin using both NMR and cryo-EM, the study presents unprecedented mechanistic insight into TcART's substrate specificity and mechanism. Using structural analysis and TIRF experiments, the authors further demonstrate that ADP-ribosylated F-actin is resistant to severing by cofilin due to a steric clash between the ADP-ribose and cofilin. This provides a mechanistic explanation for how F-actin turnover is perturbed by the toxic modification in cells.

Overall, the experiments are well-designed and the results are clearly described. The study significantly advances our understanding of a unique type of F-actin modification by a bacterial toxin, which is likely of general interest to both the actin cytoskeleton and bacterial pathogenesis fields. Below are a few specific comments and minor points which I believe should be addressed through textual revisions prior to acceptance for publication:

Comments:

1 Page 4, paragraph 2: The authors discuss the presence and importance of two hydrogen bonds between Y112 and A195. I couldn't spot them anywhere in the figures. Please show them to support the argument.

We thank the reviewer for noticing this. We have now added a figure (Supplementary Fig. 3b) illustrating the hydrogen bonds between Y112 and A195.

2 Page 8, last paragraph: The authors claimed "D263 plays a central part in the enzymatic reaction and can't be replaced by other residues". Fig. S7g clearly showed the enzymatic activity of D263N is comparable to that of WT, which contradicts the authors' claim. Please tone down the statement.

We investigated the role of D263 in the enzymatic reaction by mutating it to Asn or Gln. In both cases, we observed a significant reduction in toxicity in the yeast model (Supplementary Fig. 7c) and >10-fold decrease in the activity of the enzyme *in vitro* (Supplementary Fig. 7g). Therefore, we kindly disagree with the reviewer and believe that our data supports the claim that D263 cannot be replaced by other residues.

3 Page 9, paragraph 2: The statement "suggesting that F-actin does not prefer TcART-NAD⁺ over ligand-free TcART" contradicts Fig2b showing a clear affinity difference for E265S with/without NAD⁺ and, to a lesser extent, for WT. The presence of NAD⁺ greatly increased the affinity of E265S to F-actin. This leads to a significant concern about the proposed mechanism in which the nucleotide-free state TcART interacts with F-actin first. Either NAD or F-actin could initiate binding to TcART and trigger an induced-fit mechanism. The current data is not sufficient to support one over the other. The authors need to either provide more evidence or tone this down a bit.

Results of the cosedimentation experiments presented in Fig. 2b cannot be used to determine whether F-actin prefers TcART-NAD⁺ over ligand-free TcART because mixing TcART with F-actin in the presence of NAD⁺ leads to rapid hydrolysis of NAD⁺ and ADP-ribosylation of

actin (Fig. 2a, Supplementary Fig. 7e). In this case, cosedimentation analysis provides the affinity of the enzyme for the modified substrate and not for the non-modified F-actin. Therefore, we performed a cosedimentation assay using purified actin with the mutation that prevents ADP-ribosylation (Thr148Asn). This experiment clearly demonstrated that the presence of NAD^+ does not alter the affinity of TcART for F-actin (Supplementary Fig. 7a and b). Moreover, to address the reviewer's concerns, we also now repeated the experiment with the TcART E265S mutant (see below) and again found that the affinity of the enzyme for F-actin is independent of the presence of NAD^+ in solution. These data, together with the NMR and cryo-EM structures of TcART prove that the nucleotide-free TcART interacts with F-actin first.

Figure. Fractions of TcART that cosedimented with F-actin were quantified by densitometry and plotted against F-actin concentrations. The data are presented as mean values, the error bars correspond to standard deviations of 3 independent experiments.

4 Page 12, paragraph 3: The authors postulated “it is unlikely that the ADP-ribose modification itself is sufficient to stabilize actin-actin interactions to an extent that would lead to a toxic effect.” The clear density of ADP-ribose engages two adjacent actin subunits near the D-loop, which is a hotspot of F-actin polymerization regulation. This could support an alternative explanation in which ADP ribosylation perturbs actin dynamics by accelerating actin polymerization at the barbed end. Mannherz and colleagues (Lang, A.E. et al, 2017) demonstrated rapid actin polymerization in the presence of TccC3 modified actin. What is your comment on this point?

We thank the reviewer for this comment. As mentioned in the manuscript, when analyzing the 3D reconstruction of ADP-ribosylated F-actin, we noted that the density corresponding to the ADP-ribose appears only at a lower threshold and is not as well defined as the rest of the map indicating high flexibility. Moreover, we did not detect any specific interactions between the ADP-ribose and the D-loop when analyzing the atomic model. In addition, our colleagues have previously demonstrated that the critical concentration of polymerization of ADP-ribosylated actin is almost identical to that of native actin (Lang, A.E. et al, 2010, Lang, A.E. et al, 2017). Therefore, we suggest that the modification itself cannot sufficiently stabilize actin-actin interactions in the middle of filaments. However, we do agree with the reviewer that ADP-ribosylation could affect the subtle equilibrium at the barbed end and thus accelerate actin polymerization. We have added several sentences to the discussion with reference to Mannherz and colleagues.

In their study, TccC3 ribosylates both G-actin and F-actin, which differs from this manuscript. The authors should explain the discrepancy in the Discussion section.

Indeed, Lang et al., tested whether G- or F-actin is the substrate of TccC3. However, according to the materials and methods, our colleagues did not stabilize monomeric actin, which should have partially polymerized during the 10-minutes of the ADP-ribosylation reaction. This likely resulted in their observation (we have now added this information to the manuscript).

In contrast, in our experiments, we first performed high-speed centrifugation to remove actin aggregates and then used latrunculin A to prevent spontaneous polymerization of G-actin to truly distinguish the form of the TccC3 substrate. Consistent with the biochemical data, our high-resolution cryo-EM structure of the TccC3-F-actin complex clearly shows that the binding interface of TccC3 on F-actin is located between two consecutive actin subunits. Since this interface does not exist in G-actin, TccC3 cannot bind and modify actin monomers.

5 In Fig S7e, E265S dramatically reduces ribosylation activity. I am confused why in the structure T148 is heavily ribosylated.

In the Supplementary Fig. 7e we have presented results of **10 minutes**-long ADP-ribosylation of 1 μg (2.3 μM) F-actin by 100 ng (**0.16 μM**) of the E265S TccC3 variant. Under these conditions, we observed only a minor modification of the substrate. However, for the preparation of the cryo-EM sample, we incubated 15 μM of F-actin with **25 μM** of TccC3 for 60 minutes, which allowed us to obtain a structure with modified T148 of F-actin.

What is equally confusing is why E265S mutant didn't dissociate from F-actin after ribosylation in the presence of excessive NAD^+ in solution. A short explanation is needed.

We thank the reviewer for this question. We cannot answer this question with certainty. However, we can speculate that E265S also slows down the enzymatic reaction and prevents rapid release of the reaction product. This is now discussed in the revised manuscript.

6 Based on co-sedimentation assay, a significant portion of (>20%) wildtype TcART co-sediments with 16 μM of actin. Are there any density for WT TcART during data processing? The authors didn't indicate the concentration of actin mixed with wildtype TcART in the Methods. Presumably it is the same as that with TcART E265S.

Indeed, we forgot to report the exact concentration of F-actin mixed with WT TcART in the Methods section. We have now added this information. Actin and toxin were used in a molar ratio of 100 to 1. We did not observe density corresponding to wildtype TcART, which is expected due to the relatively low affinity of TcART for F-actin.

7 It is surprising that there is no binding of cofilin to ADP-ribosylated F-actin in the TIRF assay. I assume the filament is not fully ribosylated. Does that suggest ADP-ribosylation on T148 induces long range allosteric effect to the sites where T148 is not ribosylated? Are there structural differences between unmodified and ADP ribosylated F-actin? Lifeact interferes with cofilin severing. What is the percentage of Lifeact-F-actin used for the assay? Please add more details in the Methods.

For the TIRF microscopy experiment, we used fully ADP-ribosylated F-actin. We did not detect any structural differences between unmodified and ADP-ribosylated F-actin, except for the density for the ADP-ribose moiety.

We agree that Lifeact can inhibit cofilin binding and severing at high concentrations as shown by the Pollard lab (Courtemanche et al 2016, PMID 27159499). However, at the trace concentration used in our TIRF assay (10 nM), it only weakly decorates actin (one Lifeact molecule per 200 actin protomers based on the known affinity (2 μ M) of Lifeact for F-actin). In agreement with very weak decoration under our assay conditions, we can readily observe cofilin binding and severing in our control experiments, which use the same low Lifeact concentration (Fig. 4C, top). This is also in line with previous data from the Pollard lab that observed a marginal (20%) reduction in cofilin severing at 10nM Lifeact.

We added more details about TIRF microscopy in the Methods.

Minor points:

1. Page 7, paragraph 2: add a space between (G-) and actin.

We believe that there should be no space in “iota toxin-G-actin complex”.

2. Page 9, paragraph 2: please correct “(Fig 3a, b, 5)”.

We think that this is correct. We indeed cite figure 5 there.

3. In the Methods, please add the purification of actin T148N.

Thank you for noticing this. We now described purification of actin T148N.

4. Figure S8a and S10a: please add length parameter to the scale bar.

We now added the length parameter to scale bars.

Reviewer #3 (Remarks to the Author):

Prior studies show that the TccC3 hypervariable region ADP ribosylates actin residue T148, and that this leads to aberrant actin polymerization. Belyy et al present new data showing the structure of TccC3 hypervariable region, that the preferred substrate for the bacterial ADPr toxin TccC3 is F-actin, and that this modification prevents association and activity of depolymerizing factor cofilin.

The authors provide convincing evidence that the TccC3 hypervariable region is initially unfolded and positioned randomly inside the TcB-TcC cocoon using MAS NMR. They provide a rationale for their approach to determine the structure of TccC3 HVR, explaining why crystallography was not possible, and using NMR. They determine that the unstructured N-terminal region was not needed for toxicity, and propose it may serve as a linker to the TcB-TcC cocoon. Using informed mutagenesis, they were able to solve the structure of TccC3 HVR bound to F-actin by Cryo-EM, and test their results with mutagenesis of other key residues, showing it to differ from other RSE bacterial ARTs. Within this structure were densities for ADP ribose and nicotinamide, permitting them to investigate the most probable mechanism of ADP ribosylation. The authors also investigate how this modification leads to aberrant

polymerization, showing it hinders association for cofilin using a previously published TIRF assay to monitor labeled actin filament dynamics with labeled cofilin.

The paper is well-written and easy to read. Inclusion of experiments in the text that did not yield results provide a clear rationale for the approaches the authors take. Data are clearly presented and the movie figure is a helpful addition. Overall this is an interesting addition to the Tc toxin and bacterial ART fields.

We thank the reviewer for the very positive feedback on our manuscript.